# Federated Deep Survival Learning for Hip Fracture Risk Prediction from Pelvic X-rays in the Study of Osteoporotic Fractures

**Niklas C. Koser**[*1] iD                              NIKLAS.KOSER@INFORMATIK.UNI-KIEL.DE

**Jesse Lindstädt**[*1]

**Marten J. Finck**[*1] iD                              MAFI@INFORMATIK.UNI-KIEL.DE

**Sören Pirk**[1] iD                                    SP@INFORMATIK.UNI-KIEL.DE

**Claus-C. Glüer**[2] iD                                GLUEER@RAD.UNI-KIEL.DE

[1] *Visual Computing and Artificial Intelligence, Kiel University, Kiel, Germany*

[2] *i2Lab@SBMI, Kiel University, University Hospital Schleswig-Holstein, Kiel, Germany*

## Abstract

Hip fractures are a major cause of morbidity and mortality among older adults. Although deep learning models show promise for predicting hip fracture risk, the development of robust and generalizable models is constrained by cross-institutional data-sharing restrictions. Federated learning (FL) addresses this challenge by enabling collaborative training without exchanging raw patient data. We present the first FL framework for hip fracture risk prediction from pelvic X-rays, training a DeepSurv-based survival model across four clinical sites of the Study of Osteoporotic Fractures (SOF). We compare FedAvg, FedBN, FedProx, FedNova, and a site-specific personalization adapter in pairwise and triple-site settings. All FL strategies outperform isolated site-specific training, and the best methods match or exceed centralized pooled training, demonstrating the potential of privacy-preserving multi-center risk prediction. Code will be released upon acceptance.

**Keywords:** Federated Learning, Hip Fracture Risk, Osteoporosis, Deep Learning

## 1. Introduction

Osteoporosis is a systemic skeletal disease defined by reduced bone strength and an increased risk of fractures. Among these, hip fractures are associated with higher mortality, reduced quality of life, and an increasing burden on healthcare systems (Shen et al., 2022). Despite established approaches such as Dual Energy X-ray Absorptiometry-derived bone mineral density (BMD) and FRAX® (Kanis et al., 2007), fracture risk prediction remains limited (Koser et al., 2026). Routinely acquired pelvic X-rays may provide additional structural information and have shown considerable potential for deep learning-based risk prediction (Damm et al., 2022; Hsieh et al., 2021; Koser et al., 2026). However, deep learning models require large and diverse datasets, which are difficult to obtain in medical imaging due to privacy regulations and limited data sharing across institutions. Federated learning (FL) addresses this challenge by enabling collaborative training without sharing raw patient data. Prior federated survival work has mainly focused on clinical or other non-radiographic data (Rahman and Purushotham, 2023; Andreux et al., 2020), although imaging-based exceptions have emerged in areas such as computational pathology (Lu et al., 2022) and

---

[*] Contributed equally

multimodal PET-based survival prediction (Vo et al., 2024). In parallel, radiograph-based models for fracture risk assessment have been developed in centralized settings (Koser et al., 2026). To our knowledge, federated time-to-event prediction of future hip fracture risk from hip X-rays has not yet been reported. In this work, we (1) present the first federated framework for hip fracture risk prediction from pelvic X-rays using a DeepSurv-based model (Katzman et al., 2018) trained across four clinical sites without sharing raw patient data, and (2) compare FedAvg (McMahan et al., 2017), FedBN (Li et al., 2021), FedProx (Li et al., 2020), FedNova (Wang et al., 2020), and a site-specific PersonalHead adapter.

## 2. Methods

**Dataset and Preprocessing.** This study uses pelvic X-rays from the Study of Osteoporotic Fractures (SOF), which has a follow-up period of up to 23 years. This enables time-to-event modelling of incident hip fracture risk. In the FL setup, the four SOF clinical centres were treated as individual clients and the full dataset was divided into training, validation and test sets. Detailed cohort statistics and data splits by site are provided in Appendix 2. For preprocessing, we followed (Koser et al., 2026) and used a keypoint detection model to extract a standardized hip crop centered on the trochanteric and femoral neck region. **Risk Prediction Model.** The hip fracture risk prediction model consists of a ConvNeXt-Tiny (Liu et al., 2022) backbone (pretrained on ImageNet-1K), as well as a two-layer prediction head (FC(768→256), ReLU, Dropout, FC(256→1)). The model is trained based on the CoxPH partial log-likelihood such that the output indicates the relative risk for each patient. **Baseline Training** As reference baselines, we train the same model architecture in both centralized and local settings. In the centralized setting, a single model is trained on the pooled multi-site dataset. In the local setting, separate models are trained independently on each clinical site. These baselines provide lower and upper reference points for evaluating the performance of FL. **Federated Learning Setup.** FL is implemented in NVFlare, which handles communication, serialization, and checkpointing across sites. In each communication round, the server sends the current global model to all clients, each site performs local optimization on its private data, and the resulting updates are aggregated on the server. No raw patient data leave the local clinical center. **Aggregation and Personalization.** We compare four established FL strategies: FedAvg, FedBN, FedProx, and FedNova. FedAvg serves as the standard weighted model averaging baseline. FedBN keeps normalization parameters local to better account for site-specific domain shifts, while FedProx adds a proximal regularization term to stabilize local training under data heterogeneity. FedNova further corrects for differences in local optimization by normalizing client updates before aggregation. In addition, we evaluate a personalized variant, denoted PersonalHead, in which the ConvNeXt encoder is shared across sites while the prediction head remains site-specific. **Experimental Setup.** We evaluate all methods in pairwise and triple-site federated configurations across the four SOF centers. Model selection is performed on the 70/30% validation split, and performance is measured using the concordance index (C-index) for time-to-event prediction. Federated methods are compared against both centralized pooled training and site-specific local baselines.

## 3. Results

Table 1 summarizes test-set C-indices for all baselines and FL strategies with 95% confidence intervals from bootstrapping ($n = 2000$). The local baseline showed the lowest performance, because each center only has access to a subset of the data. Training on the pooled dataset resulted in a higher C-index of 0.741. Among the FL approaches, FedBN achieved the highest C-index of 0.774. We assume that this is due to FedBN's ability to account for differences in data distributions across medical centers. When combined with FedProx, the C-index was slightly lower at 0.772, possibly because FedProx keeps the local weights closer to the global model and thus slightly reduces the benefit of FedBN. Overall, most FL approaches reached C-indices similar to the standard FL method FedAvg. Further results, as well as a cross-site evaluation and AUCs, can be found in the Appendix 4.

Table 1: Main results on the held-out SOF test set. Values are reported as C-index with 95% confidence intervals on the independent test set.

| Method | C-index |
|---|---|
| Local baseline | **0.715** (0.692 - 0.736) |
| Pooled baseline | **0.741** (0.721 - 0.761) |
| FedAvg | **0.748** (0.692 - 0.799) |
| Best Single FL Method: FedBN | **0.774** (0.718 - 0.825) |
| Best Combined FL Method: FedBN+FedProx | **0.772** (0.722 - 0.820) |
| PersonalHead | **0.748** (0.696 - 0.798) |

## 4. Conclusion, Limitations and Future Work

Despite being limited to the SOF cohort, relatively similar site distributions, and a single held-out test set, our results are encouraging. We present the first federated framework for hip fracture risk prediction from pelvic X-rays and show that DeepSurv-based survival models can be trained across four clinical sites. Federated training matched, and in three cases exceeded, centralized pooled training while clearly outperforming local site-specific models. This supports FL as a promising approach for privacy-preserving hip fracture risk prediction. Future work will extend the framework to additional cohorts such as MrOS and integrate X-rays with clinical variables in a multimodal setting.

### Acknowledgments

This project was supported by the modular AI Imaging Pipelines (mAIPipes) Grant, Application No. 22024025 KI-Förderrichtlinie Schleswig-Holstein, Germany. The Study of Osteoporotic Fractures (SOF) is supported by National Institutes of Health funding. The National Institute on Aging (NIA) provides support under the following grant numbers: R01 AG005407, R01 AR35582, R01 AR35583, R01 AR35584, R01 AG005394, R01 AG027574, and R01 AG027576. We also would like to thank Sharmila Majumdar for her support.

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

## Appendix

### Dataset Statistics and Data Splits

Table 2: Site-wise cohort statistics and data splits for the SOF federated learning setup. Numbers are reported as sample count / incident hip fractures.

| Site | Train ($n$/evt) | Test ($n$/evt) | Mean age (years) |
|---|---|---|---|
| Baltimore | 1429 / 155 | 587 / 69 | 72.1 ± 0.690 |
| Pittsburgh | 1144 / 182 | 95 / 13 | 71.4 ± 0.690 |
| Minneapolis | 1298 / 196 | 629 / 97 | 72.3 ± 0.690 |
| Portland | 1418 / 236 | 658 / 106 | 71.5 ± 0.690 |
| Total | 5289 / 769 | 1969 / 285 | 71.8 ± 0.690 |

### Detailed Evaluation

In addition to the global model, we evaluated best-per-site (BPS) federated checkpoints selected separately for each site based on local validation performance. Since the non-federated baselines were not trained in a federated setting, no BPS model is available for these methods.

Table 3: Main results on the held-out SOF test set. Values are reported as C-index / AUC with 95% confidence intervals on the independent test set.

| Method | BPS C-index | BPS AUC | Global C-index | Global AUC |
|---|---|---|---|---|
| Local (siloed) | — | — | 0.715 (0.692 - 0.736) | 0.676 (0.652 - 0.700) |
| Central (pooled) | — | — | 0.741 (0.721 - 0.761) | 0.704 (0.681 - 0.726) |
| FedAvg | 0.744 (0.690 - 0.795) | 0.708 (0.649 - 0.763) | 0.748 (0.692 - 0.799) | 0.709 (0.649 - 0.765) |
| FedBN | 0.768 (0.711 - 0.820) | 0.731 (0.670 - 0.790) | **0.774** (0.718 - 0.825) | **0.737** (0.676 - 0.794) |
| FedProx | 0.735 (0.680 - 0.785) | 0.695 (0.635 - 0.753) | 0.740 (0.685 - 0.792) | 0.699 (0.639 - 0.756) |
| FedNova | 0.741 (0.685 - 0.794) | 0.702 (0.642 - 0.758) | 0.742 (0.686 - 0.794) | 0.704 (0.645 - 0.760) |
| FedBN+FedProx | **0.777** (0.726 - 0.826) | **0.735** (0.677 - 0.790) | 0.772 (0.722 - 0.820) | 0.731 (0.673 - 0.786) |
| FedProx+FedNova | 0.776 (0.723 - 0.826) | 0.735 (0.676 - 0.790) | 0.770 (0.716 - 0.819) | 0.730 (0.670 - 0.786) |
| FedBN+FedNova | 0.740 (0.687 - 0.791) | 0.695 (0.637 - 0.752) | 0.743 (0.690 - 0.793) | 0.699 (0.641 - 0.754) |
| FedBN+FedProx+FedNova | 0.744 (0.690 - 0.797) | 0.706 (0.646 - 0.766) | 0.747 (0.693 - 0.797) | 0.709 (0.649 - 0.766) |
| PersonalHead | 0.749 (0.697 - 0.800) | 0.709 (0.651 - 0.767) | 0.748 (0.696 - 0.798) | 0.708 (0.649 - 0.765) |

### Architecture Comparison

Table 4 reports centralized (pooled) training for all four backbone architectures on the SOF test set (3,664 patients, bootstrap $n = 2000$). ConvNeXt-Tiny achieves the highest predictive performance across both metrics. It outperforms older convolutional networks (VGG-16, DenseNet-121) as well as the heavily parameterized Vision Transformer (ViT-B/16), which likely requires substantially larger datasets to fully optimize its self-attention mechanisms. Consequently, ConvNeXt-Tiny provides the best trade-off between model complexity and accuracy, justifying its selection as the standard backbone for all federated experiments.

Table 4: Centralized SOF architecture comparison (test set, bootstrap $n=2000$, 95 % CI).

| Architecture | Params | C-index [95 % CI] | AUC [95 % CI] |
|---|---|---|---|
| ConvNeXt-Tiny | 27.8 M | **0.776** (0.757 - 0.795) | **0.731** (0.709 - 0.753) |
| VGG-16 | 14.8 M | 0.764 (0.744 - 0.784) | 0.692 (0.668 - 0.739) |
| ViT-B/16 | 86.0 M | 0.762 (0.741 - 0.784) | 0.721 (0.698 - 0.744) |
| DenseNet-121 | 7.2 M | 0.748 (0.728 - 0.770) | 0.711 (0.688 - 0.735) |

**Full Cross-Site $4 \times 4$ C-index Matrices**

All nine cross-site evaluation matrices. Rows = training site (whose local model is used); columns = evaluation site; **bold diagonal** = home evaluation. These matrices demonstrate the generalization capability of each FL strategy across different clinical centers. Off-diagonal elements reveal potential performance drops caused by inter-site domain shifts.

Table 5: SOF cross-site C-index — FedBN+FedNova (BPS Rank 1).

| | Minneapolis | Portland | Baltimore | Pittsburgh |
|---|---|---|---|---|
| Minneapolis | **0.722** | 0.741 | 0.760 | 0.755 |
| Portland | 0.721 | **0.741** | 0.758 | 0.748 |
| Baltimore | 0.720 | 0.739 | **0.760** | 0.753 |
| Pittsburgh | 0.723 | 0.742 | 0.758 | **0.747** |

Table 6: SOF cross-site C-index — FedNova (BPS Rank 2).

| | Minneapolis | Portland | Baltimore | Pittsburgh |
|---|---|---|---|---|
| Minneapolis | **0.718** | 0.733 | 0.756 | 0.748 |
| Portland | 0.719 | **0.731** | 0.758 | 0.760 |
| Baltimore | 0.720 | 0.732 | **0.762** | 0.762 |
| Pittsburgh | 0.719 | 0.732 | 0.758 | **0.762** |

Table 7: SOF cross-site C-index — FedProx (BPS Rank 3).

| | Minneapolis | Portland | Baltimore | Pittsburgh |
|---|---|---|---|---|
| Minneapolis | **0.720** | 0.722 | 0.767 | 0.750 |
| Portland | 0.718 | **0.726** | 0.765 | 0.743 |
| Baltimore | 0.720 | 0.724 | **0.768** | 0.748 |
| Pittsburgh | 0.718 | 0.729 | 0.763 | **0.767** |

Table 8: SOF cross-site C-index — FedBN+FedProx (BPS Rank 4).

| | Minneapolis | Portland | Baltimore | Pittsburgh |
|---|---|---|---|---|
| Minneapolis | **0.759** | 0.766 | 0.808 | 0.772 |
| Portland | 0.749 | **0.757** | 0.801 | 0.773 |
| Baltimore | 0.753 | 0.760 | **0.804** | 0.772 |
| Pittsburgh | 0.753 | 0.759 | 0.802 | **0.771** |

Table 9: SOF cross-site C-index — PersonalHead (BPS Rank 5).

| | Minneapolis | Portland | Baltimore | Pittsburgh |
|---|---|---|---|---|
| Minneapolis | **0.714** | 0.736 | 0.767 | 0.757 |
| Portland | 0.716 | **0.746** | 0.771 | 0.779 |
| Baltimore | 0.721 | 0.738 | **0.772** | 0.761 |
| Pittsburgh | 0.719 | 0.737 | 0.772 | **0.769** |

Table 10: SOF cross-site C-index — FedProx+FedNova (BPS Rank 6).

| | Minneapolis | Portland | Baltimore | Pittsburgh |
|---|---|---|---|---|
| Minneapolis | **0.755** | 0.758 | 0.805 | 0.765 |
| Portland | 0.764 | **0.770** | 0.812 | 0.768 |
| Baltimore | 0.751 | 0.754 | **0.801** | 0.751 |
| Pittsburgh | 0.747 | 0.748 | 0.797 | **0.767** |

Table 11: SOF cross-site C-index — FedAvg (BPS Rank 7).

|  | Minneapolis | Portland | Baltimore | Pittsburgh |
|---|---|---|---|---|
| Minneapolis | **0.709** | 0.738 | 0.771 | 0.784 |
| Portland | 0.710 | **0.739** | 0.770 | 0.767 |
| Baltimore | 0.711 | 0.738 | **0.771** | 0.784 |
| Pittsburgh | 0.709 | 0.739 | 0.767 | **0.760** |

Table 12: SOF cross-site C-index — FedBN+FedProx+FedNova (BPS Rank 8).

|  | Minneapolis | Portland | Baltimore | Pittsburgh |
|---|---|---|---|---|
| Minneapolis | **0.710** | 0.734 | 0.772 | 0.769 |
| Portland | 0.711 | **0.730** | 0.776 | 0.772 |
| Baltimore | 0.709 | 0.730 | **0.772** | 0.770 |
| Pittsburgh | 0.711 | 0.732 | 0.773 | **0.777** |

Table 13: SOF cross-site C-index — FedBN (BPS Rank 9).

|  | Minneapolis | Portland | Baltimore | Pittsburgh |
|---|---|---|---|---|
| Minneapolis | **0.786** | 0.789 | 0.812 | 0.733 |
| Portland | 0.776 | **0.779** | 0.810 | 0.755 |
| Baltimore | 0.764 | 0.768 | **0.800** | 0.727 |
| Pittsburgh | 0.763 | 0.767 | 0.799 | **0.755** |

