# OpenReview forum: "Federated Deep Survival Learning for Hip Fracture Risk Prediction from Pelvic X-rays in the Study of Osteoporotic Fractures"
_MIDL.io/2026/Short_Papers — MIDL 2026 - Short Papers Poster_

### Official Review · Reviewer_phTM · 2026-05-03
**Federated Deep Survival Learning for Hip Fracture Risk Prediction from Pelvic X-rays**

**Rating:** 3
**Confidence:** 4

**Review:**

This short paper addresses a clinically relevant problem, and the federated learning setup is well motivated. The contribution is primarily applied—combining existing FL strategies with an established survival model on a well-known cohort—but the combination in this clinical context is of interest and fits the scope of a short paper. However, several methodological gaps limit the strength of the conclusions and the validity of some claims.

**Summary:**

This short paper presents a federated learning (FL) framework for hip fracture risk prediction from pelvic X-rays. The authors compare multiple FL strategies (FedAvg, FedBN, FedProx, FedNova, and a PersonalHead adapter) on the Study of Osteoporotic Fractures (SOF) dataset. All FL approaches outperform isolated site-specific training, and some slightly outperform centralized pooled training.

**Strengths:**

- Clinically relevant problem with a well-motivated privacy-preserving framing.
- Thorough comparison of multiple FL strategies (FedAvg, FedBN, FedProx, FedNova, and a PersonalHead adapter).
- Federated results outperform centralised training.
- Clearly written and easy to follow.

**Weaknesses:**

- Inter-site heterogeneity of the Study of Osteoporotic Fractures is likely lower than in real-world deployments, limiting generalisability.
- The authors claim privacy preservation without discussing differential privacy or secure aggregation, which should at minimum be acknowledged as a limitation.
- The number of local epochs per communication round and total global rounds are not reported. These are critical FL hyper-parameters.
- It is unclear whether the validation set used for model selection is centralised or local per site. A centralised validation set would be unavailable in a true FL deployment, potentially introducing an unrealistic advantage in model selection.

**Justification Of Rating:**

This short paper addresses a relevant clinical problem and presents encouraging FL results, but several methodological gaps limit the conclusions. The privacy claim is overstated without discussion of differential privacy, critical training hyper-parameters are missing, and the validation setup may not reflect true FL conditions. Overall, the work is of interest to the MIDL community but requires clarification on experimental setting details.

---

### Decision · Program_Chairs · 2026-05-08

Accept (Poster)